# Remaining Useful Life Prediction Method of PEM Fuel Cells Based on a Hybrid Model

Qiancheng Tian [1,2], Haitao Chen [1,2], Shuai Ding [1,2], Lei Shu [3,4], Lei Wang [1,2] and Jun Huang [1,2,*]

1   Shanghai Institute of Space Power-Sources, Shanghai 200245, China; tqc010901@163.com (Q.T.); haitaochen2014@163.com (H.C.); dingshuai811@163.com (S.D.)
2   State Key Laboratory of Space Power Sources, Shanghai 200245, China
3   Beijing Microelectronics Technology Institute, Beijing 100076, China; shulei@pku.edu.cn
4   School of Integrated Circuit, Peking University, Beijing 100871, China
*   Correspondence: huju1981@163.com

**Abstract:** To predict the remaining useful life (RUL) of the proton exchange membrane fuel cell (PEMFC) in advance, a prediction method based on the voltage recovery model and Bayesian optimization of a multi-kernel relevance vector machine (MK-RVM) is proposed in this paper. First, the empirical mode decomposition (EMD) method was used to preprocess the data, and then MK-RVM was used to train the model. Next, the Bayesian optimization algorithm was used to optimize the weight coefficient of the kernel function to complete the parameter update of the prediction model, and the voltage recovery model was added to the prediction model to realize the rapid and accurate prediction of the RUL of PEMFC. Finally, the method proposed in this paper was applied to the open data set of PEMFC provided by Fuel Cell Laboratory (FCLAB), and the prediction accuracy of RUL for PEMFC was obtained by 95.35%, indicating that this method had good generalization ability and verified the accuracy of the method when predicting the RUL of PEMFC. The realization of long-term projections for PEMFC RUL not only improves the useful life, reliability, and safety of PEMFC but also reduces operating costs and downtime.

**Keywords:** remaining useful life; empirical mode decomposition; Bayesian optimization algorithm; multi-kernel relevance vector machine; pem fuel cell

## 1. Introduction

As a leading technology of clean and renewable energy, PEMFC has the advantage of high energy conversion efficiency and less environmental pollution compared with traditional internal combustion engines. At present, PEMFC has been developing rapidly in the fields of distributed power generation, power networks, fixed power generation, and automotive energy, which is the key direction of future new energy development and has a good market prospect. Despite this, PEMFC still has the problems of short useful life and high production costs, which seriously affect the commercial application and popularization of PEMFC. In addition to breakthrough innovations in electrochemical materials, RUL prediction research is also one a feasible method to improve the life of PEMFC because it can predict the life state of the reactor in advance and, thus, improve useful life. Carrying out RUL predictions based on PEMFC has gradually become a hot topic for researchers.

To promote the commercial application of PEMFC, the US Department of Energy (DOE) has formulated the corresponding standard, where the life of PEMFC should reach 8000 h in 2025, and the maximum output power of the reactor should ensure that the maximum output power cannot be less than 90% of the rated output power [1]. At present, according to relevant research in the literature both at home and abroad, the model-driven method [2–6], data-driven method [7–12], and hybrid model [13–17] method are the main methods for the RUL prediction of PEMFC.

The model-driven method uses the empirical model or the mechanism model for the RUL of PEMFC [18,19]. Koltsova [20] proposed an electrochemical reaction area decay mechanism model to predict the RUL of the reactor through the decay of the electrochemical reaction area. Since it is difficult to directly obtain the electrochemical reaction area during the use of fuel cells, this method is only suitable for laboratory research. Robin [21] improved on this basis and established the Pt catalyst's dissolution mechanism model and the voltage decay's semi-mechanism model. The fusion of these two mechanism models effectively improved the prediction accuracy of the RUL of fuel cells. Its advantage is that data requirements are not high, with high accuracy, which can not only observe the degradation process inside the reactor but can also observe the change in the key parameters of the reactor. However, it is very complicated to establish an accurate mechanism model, so it is necessary to be familiar with the degradation mechanism of the reactor and have a strong modeling ability. There are still many mechanisms inside the reactor that have not been explored clearly [22,23].

The data-driven method is to realize the RUL prediction of PEMFC by monitoring the status of the reactor system [24]. Silva [25] proposed a PEMFC degradation prediction method based on an adaptive neuro-fuzzy inference system (ANFIS), which used the output voltage value to predict the aging degree of the fuel cell system. The voltage signal was divided into two parts: normal operation and external disturbances to reduce the prediction errors caused by external disturbances. This method was evaluated by predicting the output voltage variation in the fuel cell reactor under constant operating conditions in a long-term experiment. Wu [26] proposed a PEMFC performance degradation prediction method based on adaptive RVM. During the training period, to obtain the behavior characteristics of PEMFC aging data, the design matrix was extended by attaching non-kernel columns, and the adaptive kernel width determination algorithm was used to make the training or learning process more intelligent and effective. Adaptive RVM was trained and tested using experimental voltage aging data from two different PEMFC reactors (1.2 kW Ballard PEMFC and 8 kW PM200 PEMFC). This data-driven approach relied on the nature of the data, and the establishment of the model required a large amount of data. The prediction accuracy of the model no longer depended solely on the merits of the algorithm but more on the merits of the experimental data. Therefore, data-driven methods were mostly applied to short-term prediction.

The hybrid model method combines the advantages of various models to improve the prediction accuracy of the RUL of the reactor to a greater extent [27]. Cheng [28] proposed a fuel cell RUL prediction method based on the least square support vector machine (LSSVM) and regularized particle filter (RPF). First, LSSVM was used to realize the preliminary prediction of PEMFC. Then, the predicted voltage value of LSSVM was used as the new system observation value by RPF, and the uncertainty characteristic of the predicted result was output in the form of RUL probability distribution, which improved the prediction accuracy of RUL in the reactor. Liu [29] proposed a short-term prediction method for fuel cells based on the group method of data handling (GMDH) and wavelet analysis (WA). The coefficient of determination (R2), mean absolute percent error (MAPE), and root mean square error (RMSE) were used. Two sets of PEMFC aging experimental data sets were used to verify the effectiveness of the method under different current load conditions, and a more accurate prediction accuracy was obtained.

Although the method based on the mechanism model has high prediction accuracy, it is often difficult to obtain an accurate mechanism model. The data-driven method overcomes the difficulty of obtaining the mechanism model, but it requires a large number of standard data sets for training, and the quality of the data sets has a great impact on the accuracy of the prediction [30]. The method based on the hybrid model combines the mechanism model and the data drive [31], takes the long and avoids the short, solves the life prediction problem of both effectively [32], and improves the accuracy of prediction.

Therefore, according to the life performance characteristics of PEMFC, an RUL prediction method based on a hybrid model is proposed in this paper, which can realize the

long-term accurate prediction of RUL. In this paper, the original data set was analyzed and processed. Voltage was determined as the performance degradation index of the reactor by Person correlation analysis, and voltage data were pre-processed by the EMD denoising method. Then, a RUL prediction model based on the hybrid method was built, and the weight coefficient of the kernel function of MK-RVM was self-optimized by the Bayesian optimization algorithm. After adding the voltage recovery model, the RUL of PEMFC was predicted and estimated. Finally, the accuracy and feasibility of the proposed method were verified using the public data set provided by FCLAB.

## 2. Data Set Analysis and Preprocessing

### 2.1. PEMFC Experimental Data Set

In this paper, the data set FC1 of the PEMFC reactor experiment published by FCLAB Laboratory in the IEEE PHM 2014 Data Challenge was selected [33]. FCLAB is the most advanced and authoritative fuel cell academic research institution in Europe, with a total of about 200 faculty and research members in the laboratory, which is the largest fuel cell academic research group in Europe. The data selected for this project were from a reactor of five single batteries with an active area of 100 $cm^2$. Operating conditions of PEMFC: temperature was controlled at about 60 °C, the load current was controlled at 70A, and relative humidity was controlled at about 50%. The data set included multi-dimensional data such as reactor voltage, temperatures of hydrogen, velocity of hydrogen, and humidity of air. Part of the data set is shown in Figure 1.

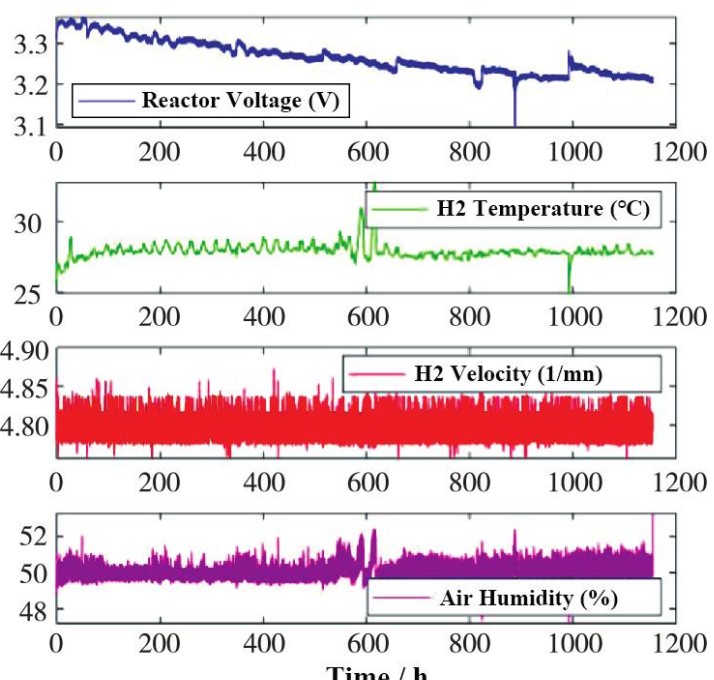

**Figure 1.** Part of the data set of PEMFC.

### 2.2. PEMFC Performance Degradation Index

PEMFC itself has the characteristics of nonlinear and time variation, while the system has multiple inputs and multiple outputs, which is a typical complex nonlinear control object [34]. These variables that can be monitored include gas flow, pressure, temperature, output voltage, output current, and output power. The PEMFC data set used in this paper had as many as 24 dimensions, as shown in Table 1.

**Table 1.** Data set parameter of PEMFC.

| Parameter | Physical Significance |
| --- | --- |
| Time | Aging time (h) |
| U1–U5, Ut | Single cells and stack voltage (V) |
| J, I | Current density (A/cm$^2$), Current (A) |
| TinH2, ToutH2 | Inlet and Outlet temperatures of H$_2$ (°C) |
| TinAIR, ToutAIR | Inlet and Outlet temperatures of Air (°C) |
| TinWAT, ToutWAT | Inlet and Outlet temperatures of cooling Water (°C) |
| PinAIR, PoutAIR | Inlet and Outlet Pressure of Air (mbara) |
| PinH2, PoutH2 | Inlet and Outlet Pressure of H$_2$ (mbara) |
| DinH2, DoutH2 | Inlet and Outlet flow rate of H$_2$ (L/min) |
| DinAIR, DoutAIR | Inlet and Outlet flow rate of Air (L/min) |
| DWAT | Flow rate of cooling water (L/min) |
| HrAIRFC | Inlet Hygrometry(Air)—estimated (%) |

To clarify the relationship between the data variables of each dimension and fuel cell performance degradation and determine the index that could best represent the performance degradation of the reactor, Person correlation analysis was carried out on the data of each dimension in the data set, and the correlation matrix between variables is shown in Figure 2.

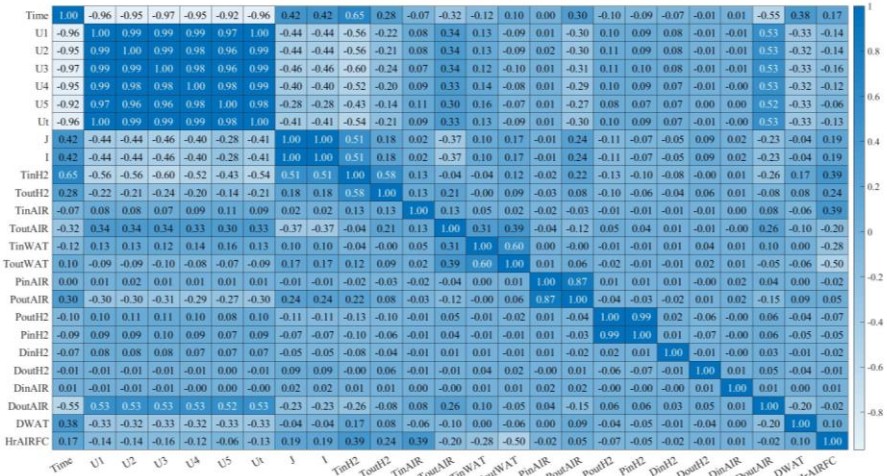

**Figure 2.** Data set variable correlation matrix diagram.

By observing the correlation coefficient between the variables in Figure 2, it can be found that the correlation coefficient between the reactor output voltage and time was about −0.95, showing an obvious negative correlation. In the RUL prediction research of PEMFC, because the output voltage of the reactor was the most easily obtained data, most research methods measured the degradation of the reactor performance through the attenuation of the output voltage of the reactor and took the output voltage of the reactor as the indicator of the performance degradation of PEMFC. Therefore, the output voltage of PEMFC was also taken as the performance degradation index of the reactor.

*2.3. EMD Denoising*

Through correlation analysis, the output voltage was determined as the performance degradation index, but there were 143,862 original output voltage data, the fluctuation between adjacent output voltage data was too small, and the calculation time required for model training was too long; therefore, it was necessary to sample the original data at equal time intervals. Considering the stable operation of the reactor for 1154 h, too short a sampling interval increased the calculation burden and affected the speed predicted of the RUL. If the sampling interval was too long, important pile degradation trend information

was lost, which affected the accuracy of RUL prediction. Therefore, 1 h was selected as the sampling interval. Because the original voltage data contained a lot of noise and voltage spikes, if these abnormal deviations are not dealt with, it produces a large calculation error in the training and prediction of the model.

EMD performs signal decomposition according to the time scale characteristics of data without presetting any basis function and overcomes the problem that the basic function has no adaptability [35]. EMD is a processing method that stabilizes non-stationary signals. It decomposes the fluctuations and trends of different scales in the signal step by step to produce a series of data series with different characteristic scales [36]. Each series is called an intrinsic mode function (IMF). In EMD, it is assumed that any signal can be decomposed into several linear or nonlinear IMF components; the local number of zeros is the same as the number of extreme values; the upper and lower envelope is locally symmetric around the time axis.

EMD needs to first find all the extreme points of the signal, connect the local maximum points into the upper envelope through the cubic spline curve, and connect the local minimum points into the lower envelope, the upper and lower envelope contains all the data points and finds the average value of the upper and lower envelope.

$$m_1(t) = \frac{e_{\max}(t) + e_{\min}(t)}{2} \tag{1}$$

Among them, $e_{\max}(t)$ is the upper envelope and $e_{\min}(t)$ is the lower envelope.

The original signal $x(t)$ is sieved, and the original signal subtracts the mean envelope to obtain the intermediate signal $C_1(t)$.

$$C_1(t) = x(t) - m_1(t) \tag{2}$$

To determine whether the middle signal $C_1(t)$ meets the two conditions of IMF: the number of extreme points and the number of zero points in the entire time course is equal to or at most one difference. At any time, the average value of the upper envelope formed by local maximum points and the lower envelope formed by local minimum points is zero; that is, the upper and lower envelope are locally symmetric concerning the time axis. If so, the signal is an IMF weight. If not, the above steps are repeated until the decomposed signal meets the IMF condition after K times to obtain the first IMF component $I_1(t)$ of the original signal.

$$C_{k-1}(t) - m_k(t) = C_k(t)I_1(t) \tag{3}$$

$$r_1(t) = x(t) - I_1(t) \tag{4}$$

Among them, $C_{k-1}(t)$ and $C_k(t)$ are intermediate signals after decomposition (K−1) and K times, $m_k(t)$ is the mean value of the envelope after K decomposition, $I_1(t)$ represents the IMF component of the highest frequency in the original signal $x(t)$, and the remaining component $r_1(t)$ is obtained by subtracting $I_1(t)$ from the original signal $x(t)$.

The second IMF component $I_2(t)$ can be obtained by screening $r_1(t)$, and the remaining component $r_2(t)$ can be obtained by subtracting $I_2(t)$ from $r_1(t)$.

$$r_2(t) = r_1(t) - I_2(t) \tag{5}$$

Therefore, until the last remaining component, $r_n(t)$ can no longer be decomposed. After n iterations, $r_n(t)$ becomes a monotone function, and the sum of all IMF components and the remaining components is the original signal $x(t)$.

EMD has obvious advantages in processing non-stationary and nonlinear data and is suitable for analyzing nonlinear and non-stationary signal sequences with high signal-to-noise ratios. Therefore, this paper used the EMD method to denoise the sampled output voltage data and the output voltage data after EMD denoising, as shown in Figure 3.

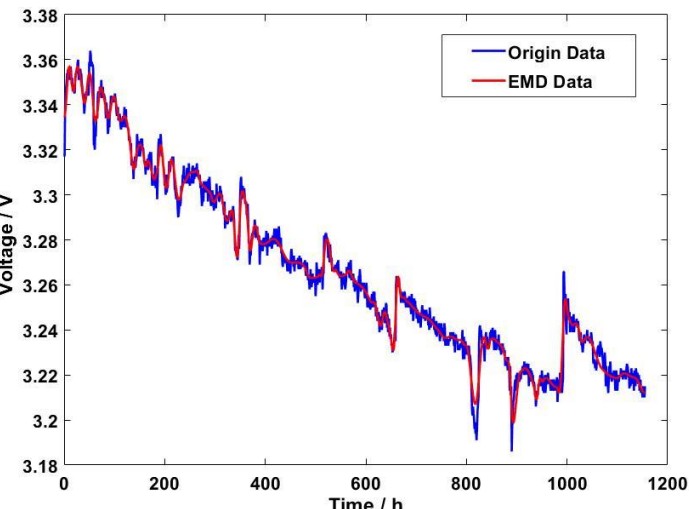

**Figure 3.** Output voltage data after EMD denoising.

## 3. RUL Prediction Method of PEM Fuel Cell Based on Hybrid Model

As the main force of clean energy, fuel cells have higher requirements for the detection and management of their health status. To further improve the prediction accuracy of fuel cells, this paper took voltage as a health indicator and proposed a method for predicting the RUL of PEMFC based on the hybrid model.

### 3.1. Prediction Framework

The overall framework of the prediction method is shown in Figure 4 below.

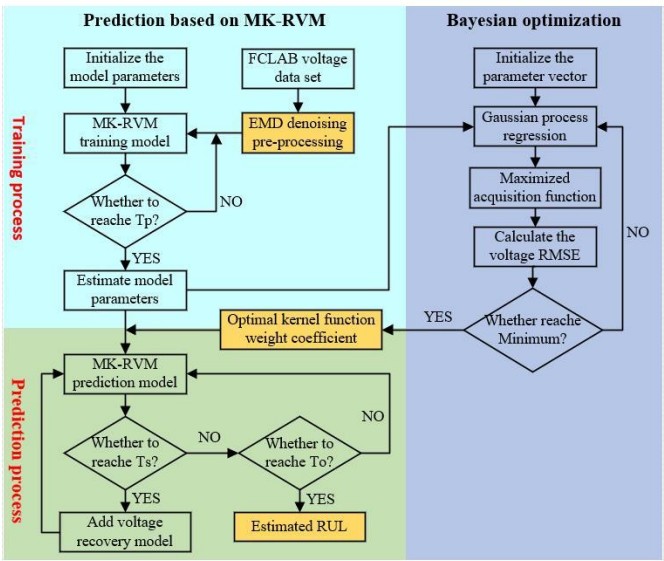

**Figure 4.** The framework of overall prediction methods.

The overall prediction process is as follows:

Step 1: Through correlation analysis, the output voltage of the PEMFC is selected as the performance degradation index of the reactor.

Step 2: The original voltage data are denoised by EMD. The denoised data are bound by the prediction starting point $T_P$ and divided into training data sets and test data sets.

Step 3: The trained data after data preprocessing into the training model of MK-RVM are trained and judge whether the RUL prediction starting point is reached. If the prediction starting point is reached, the model parameters of MK-RVM are estimated; otherwise, model training is continued.

Step 4: The model parameters of MK-RVM are optimized by Bayesian, and the model parameters are updated by Gaussian process regression and the maximized acquisition function.

Step 5: The RMSE of the reactor predicted voltage is taken as the objective function of Bayesian optimization. If RMSE reaches the minimum, the optimal kernel function weight coefficient is obtained. Otherwise, Step 4 is repeated, continuing the updating of model parameters.

Step 6: Using the weight coefficient of the optimal kernel function obtained by Bayesian optimization, the MK-RVM prediction model is updated.

Step 7: It is determined whether the start–stop time point $T_S$ of the reactor is reached. If the start–stop time point is reached, the voltage recovery model is added to the MK-RVM prediction model. Otherwise, the RUL of the PEMFC is predicted directly.

Step 8: It is then determined whether the predicted voltage value of MK-RVM reaches the threshold $T_O$ of RUL. If it reaches the voltage value of the RUL of PEMFC, the predicted RUL of the reactor is output. Otherwise, the voltage value of the PEMFC continues to be predicted.

Step 9: The uncertainty expression of RUL is realized through repeated testing to reduce the contingency of prediction and improve the generalization ability of the model.

### 3.2. MK-RVM Model

Multi-kernel learning based on RVM combines kernel functions with different characteristics to obtain the advantages of multiple kernel functions and can combine with the characteristics of global and local kernel functions when dealing with complex data to improve the learning and generalization ability of RVM [37].

The multi-kernel function $K_5(x,z)$ is the linear combination of the linear kernel function $K_1(x,z)$, Gaussian kernel function $K_2(x,z)$, polynomial kernel function $K_3(x,z)$, and Sigmoid kernel function $K_4(x,z)$, which is used to describe the global and local trend of battery capacity degradation. Its mathematical expression is shown in Equation (6):

$$K_5(x,z) = w_1 K_1(x,z) + w_2 K_2(x,z) + w_3 K_3(x,z) + w_4 K_4(x,z) \tag{6}$$

Among them, $w_1$, $w_2$, $w_3$ and $w_4$ are the weight, which is also the key of MK-RVM. Since there is no reasonable and universal criterion for setting the weight coefficient of the kernel function, it is often determined based on empirical selection, experimental comparison, large-scale search, or cross-verification method. Therefore, this paper introduces a Bayesian optimization algorithm, which takes the minimum root mean square error (RMSE) as the optimization objective and realizes the parameter self-optimization for the weight coefficient of the kernel function.

$$RMSE = \sqrt{\frac{1}{n}\sum_{i=1}^{n}\left(x(i) - \overline{x}(i)\right)^2} \tag{7}$$

Among them, $x(i)$ represents the actual voltage of the reactor and $\overline{x}(i)$ represents the predicted voltage of the reactor.

### 3.3. Bayesian Optimization Algorithm

The Bayesian optimization algorithm belongs to the black box optimization algorithm [38], which updates the posterior probability distribution based on the known observation points and the prior probability distribution of the objective function to ensure the optimal weight coefficient of the kernel function. Bayesian optimization goals are defined as:

$$x_{\min} = \mathrm{argmin}_{x\in x} f(x) \tag{8}$$

Among them, $x_{\min}$ is the final result of parameter optimization, and $f(x)$ is the objective function to be optimized.

The parameter to be optimized is set as $X = \{x_1, x_2, \ldots, x_n\}$, After Bayesian optimization iteration, the data set is $D_t = \{(x_1, f(x_1)), (x_2, f(x_2)), \ldots, (x_n, f(x_n))\}$. It is supposed that the observation points of the Gauss process obey the Gaussian distribution as follows:

$$f(x_{1:n}) \sim GP(\mu(x_{1:n}), \sum(x_{1:n}, x_{1:n})) \tag{9}$$

Among them, $\sum(x_{1:n}, x_{1:n})$ is the covariance matrix:

$$\sum(x_{1:n}, x_{1:n}) = \begin{bmatrix} k(x_1, x_1) & \cdots & k(x_1, x_n) \\ \vdots & \ddots & \vdots \\ k(x_n, x_1) & \cdots & k(x_n, x_n) \end{bmatrix} \tag{10}$$

According to Bayes' theorem:

$$P(f(x_{n+1})|f(x_{1:n})) \propto P(f(x_{1:n})|f(x_{n+1}))P(f(x_{n+1})) \tag{11}$$

Continuous iterative updates $x_{\min} = x_{t+1}$, ultimately ensure optimal parameters. The Algorithm 1 flow is as follows:

---

**Algorithm 1:** Bayesian Optimization

---

**INPUT:** objective function $f(x)$, collection function $\alpha$;
**OUTPUT:** parameter vector $x^*$;

1: Initialize parameter vector;
2: **for** $t = 1,2,\ldots,$T do;
3:　　Maximize the $\alpha$ to obtain the next evaluation point: $x_{t+1} = \arg x \epsilon X \max \alpha(x|D)$;
4:　　Evaluate the objective function value $y_{t+1} = f(x_{t+1}) + \epsilon_{t+1}$;
5:　　Integrate data: $D_{t+1} = D \cup (x_{t+1}, y_{t+1})$, and update the probabilistic proxy model;
6: End.

---

### 3.4. Voltage Recovery Model

When conducting fuel cell experiments, FCLAB not only needs to monitor the output voltage and working condition of the reactor in real time but also needs to measure the polarization curve and electrochemical impedance spectrum [39]. Therefore, the reactor's start–stop operation is carried out at regular intervals during the experiment. After the reactor stops running, the polarization curve and electrochemical impedance spectrum of the reactor are measured. After the measurement, the reactor is turned on to enter the normal operation state. The reactor's start–stop operation time data provided by FCLAB are shown in Table 2.

**Table 2.** Start–stop operation time of PEMFC.

| Start–Stop Number | 1 | 2 | 3 | 4 | 5 | 6 | 7 |
|---|---|---|---|---|---|---|---|
| Time/h | 48 | 185 | 348 | 515 | 658 | 823 | 991 |

Since the time interval of the FCLAB start–stop reactor is relatively fixed, the corresponding voltage recovery model can be established according to the voltage recovery degree of the start–stop time point in the training data [40] and the voltage recovery prediction of the predicted data start–stop point can be realized. According to the start–stop voltage recovery data of FCLAB, this paper chose the double-exponential empirical model as the start–stop voltage recovery model of the reactor, as shown in Formula (12).

$$x_c = x_k + \beta_1 \cdot \exp(\beta_2 \cdot t_k) + \beta_3 \cdot \exp(\beta_4 \cdot t_k) \tag{12}$$

Among them, $c$ is the start–stop operation time, which is the model parameter.

The comparison between the recovery amplitude of the model at the start–stop time point and the actual result is shown in Figure 5.

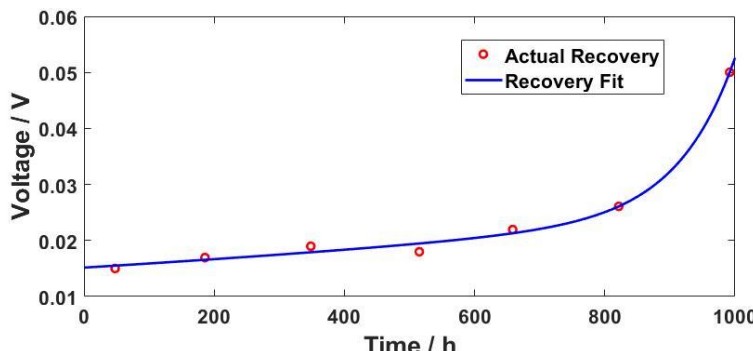

**Figure 5.** Dual exponential voltage recovery model fitting.

## 4. Experiment and Discussion

### 4.1. RUL

FCLAB provides limited experimental data on the fuel cell life, so this paper chose 3.22 V voltage as the failure threshold of the reactor; that is, 95.9% of the initial total voltage, and the failure time of the reactor was 808 h. In addition, this paper chose to set the prediction starting point at 550 h, with training data as [0, 550 h], test data as [551, 1154], and the corresponding RUL time as 258 h.

In the process of the experiment, the data were first input to MK-RVM for training, and then the RMSE of voltage prediction was used as the objective function of Bayesian optimization, and the weight coefficient of the kernel function was self-optimized by the Bayesian optimization algorithm. With the increase in the number of iterations, when the optimal value of the kernel function approached and tended to be stable, the optimal weight coefficient of the kernel function was obtained. The optimization process is shown in Figure 6.

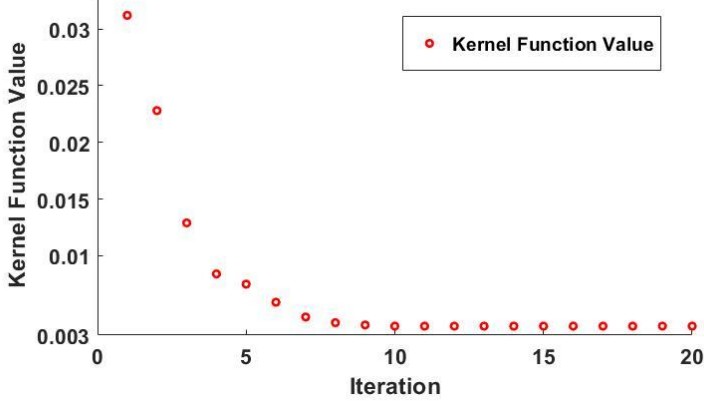

**Figure 6.** The optimization process of Bayesian optimization.

In the RUL prediction stage, the optimal kernel function weight coefficient was substituted into the prediction model. For the RUL prediction of PEMFC, the methods of MK-RVM, Bayesian optimization MK-RVM (BO-MK-RVM), and Bayesian optimization MK-RVM of voltage recovery model (VR-BO-MK-RVM) were carried out, respectively, and the prediction results are shown in Figure 7. As can be seen from the figure, the Bayesian optimization algorithm significantly improved the capture of the global voltage decay trend, and the voltage recovery model further improved the capture of the local voltage rise trend during the start and stop of the reactor. The combination of MK-RVM, the Bayesian

optimization algorithm, and the voltage recovery model effectively improved the accuracy of RUL prediction in the reactor.

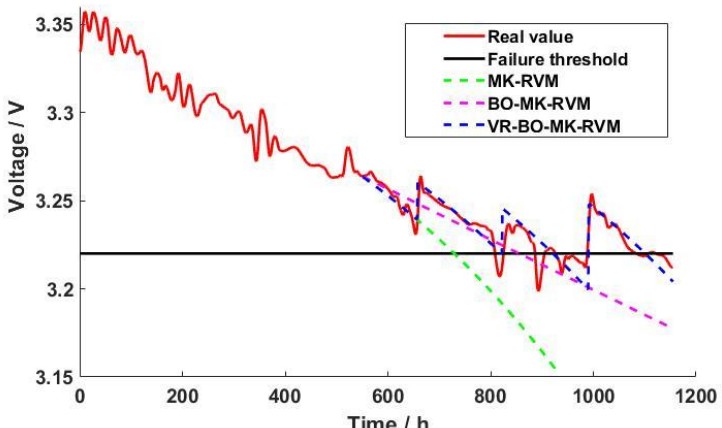

**Figure 7.** The prediction results.

The uncertainty expression of the predicted result can better guide the RUL prediction of the reactor than a single estimated result. To avoid the contingency of the prediction results of MK-RVM, the confidence of these prediction results can be verified by the repeated prediction of the model many times, and the confidence interval with a 95% significance level was added to the parameter estimation process and prediction process. The RUL confidence interval of VR-BO-MK-RVM is shown in Figure 8.

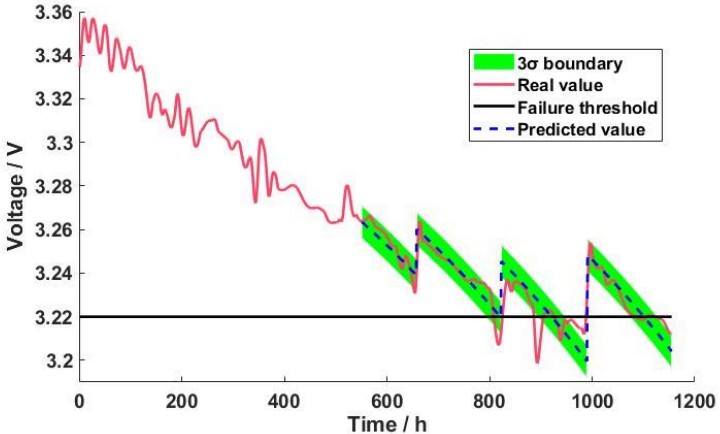

**Figure 8.** RUL confidence interval of VR-BO-MK-RVM.

Adding the voltage recovery model of reactor start–stop can effectively improve the accuracy of RUL prediction and help capture the trend of reactor voltage decay. The subsequent experimental analysis only considers the prediction results under the addition of a recovery model.

### 4.2. Prediction Result Analysis

To evaluate the reliability and validity of RUL prediction results, this paper selected the mean square error (MAE), RMSE, and relative accuracy (RA) as a model performance evaluation index, where MAE and RMSE mainly reflect the overall deviation between the predicted value and the true value, and RA is the relative prediction accuracy of RUL.

$$MAE = \frac{1}{n}\sum_{i=1}^{n}|x(i) - \overline{x}(i)| \tag{13}$$

$$RA = (1 - \frac{|R_{real} - R_{pre}|}{R_{real}}) \times 100\% \tag{14}$$

Among them, $x(i)$ is the actual voltage value of the reactor, $\overline{x}(i)$ is the predicted voltage value of the reactor, $R_{real}$ is the actual RUL of the reactor, and $R_{pre}$ is the predicted RUL of the reactor.

The detailed prediction results are shown in Table 3. When the predicted starting point was set to 550 h, the predicted and actual RUL values were 270 h and 258 h, respectively. The addition of the Bayesian optimization algorithm and voltage recovery model greatly improved the prediction accuracy of the MK-RVM method, which was conducive to the accurate prediction of PEMFC's long-term operation.

**Table 3.** The detailed prediction results.

| Algorithm | MAE | RMSE | RA | Confidence Interval |
|---|---|---|---|---|
| MK-RVM | 0.0619 | 0.0810 | 69.38% | 214 h |
| BO-MK-RVM | 0.0162 | 0.0218 | 81.78% | 106 h |
| VR-BO-MK-RVM | 0.0048 | 0.0069 | 95.35% | 29 h |

It can be seen from Table 3 that the relative accuracy of PEMFC RUL prediction was only 69.38% using only the MK-RVM method. With the addition of the Bayesian optimization algorithm, the relative accuracy of prediction was improved to 81.78%. After adding the voltage recovery model, the relative accuracy of prediction reached 95.35%, which is better than that of the LSSVM-RPF method [28].

*4.3. Discussion*

The RUL of PEMFC is difficult to predict with accuracy for three main reasons. On the one hand, PEMFC has a voltage recovery effect after the start–stop operation. For example, in the FCLAB data used in this paper, the voltage at 658 h was close to the failure threshold. If there is a large deviation in the predicted result at this time, it might be judged as the failure point, but the actual failure time point is delayed due to the voltage recovery effect caused by the start–stop operation. Therefore, the actual RUL should be considered from a global perspective.

On the other hand, PEMFC is a nonlinear complex system coupled with multiple physical fields, and many factors affect its performance and life. At present, the water, steam, and thermal management of PEMFC are not very mature, especially with the increase in the output power of the reactor, where the reactor is prone to flooding, film drying, and other failures. Sometimes, the fault situation inside the reactor is unpredictable, which affects its performance and life more or less and increases the difficulty of RUL prediction for PEMFC; therefore, it is very necessary to carry out fast online RUL prediction.

Last but not least, the complete life cycle data of PEMFC is less, and the actual operation data under various working conditions are lacking. It is difficult to accurately predict the RUL of PEMFC only by relying on a single data-driven and mechanistic model method. Only by integrating the data-driven and mechanistic model can the rapid and accurate prediction of PEMFC be achieved. The RUL prediction method based on a hybrid model needs further research.

**5. Conclusions**

In this paper, according to the PEMFC data set provided by FCLAB, a prediction method based on the voltage recovery model and Bayesian optimization MK-RVM was proposed to predict the RUL of PEMFC. In the whole prediction framework, the EMD denoising of the training data was first carried out, then MK-RVM was used for model training, and then the Bayesian optimization algorithm was adopted to realize the parameter self-optimization of the weight coefficient of the kernel function, and then the optimal

weight coefficient of the kernel function was updated to the prediction model, and then the voltage recovery model was added to the prediction model. Finally, the fuel cell data set provided by FCLAB was used to verify the accuracy of the method. The prediction accuracy of the RUL of PEMFC greatly improved, and the prediction accuracy was as high as 95.35%. This method can realize the long-term accurate prediction of PEMFC RUL, not only improving the useful life, reliability, and safety of PEMFC but also reducing operating costs and downtime. It has great practical value and provides a new way of performing fuel cell life prediction.

**Author Contributions:** Conceptualization, H.C. and J.H.; methodology, Q.T.; software, Q.T.; validation, S.D., L.W. and L.S.; formal analysis, H.C. and L.W.; investigation, H.C.; resources, L.S. and S.D.; data curation, S.D. and L.S.; writing—original draft preparation, Q.T.; writing—review and editing, Q.T.; visualization, L.W.; supervision, J.H.; project administration, J.H.; funding acquisition, L.S. All authors have read and agreed to the published version of the manuscript.

**Funding:** This research was supported in part by the National Natural Science Foundation of China under Grant 62204019.

**Data Availability Statement:** Simulation data are available on request.

**Conflicts of Interest:** The authors declare no conflict of interest. The funders had no role in the design of the study; in the collection, analyses, or interpretation of data; in the writing of the manuscript; or in the decision to publish the results.

## Abbreviations

| | |
|---|---|
| RUL | Remaining Useful Life |
| PEMFC | Proton Exchange Membrane Fuel Cell |
| MK-RVM | Multi-kernel Relevance Vector Machine |
| EMD | Empirical Mode Decomposition |
| FCLAB | Fuel Cell Laboratory |
| DOE | Department of Energy |
| ANFIS | Adaptive Neuro-fuzzy Inference System |
| LSSVM | Least Square Support Vector Machine |
| RPF | Regularized Particle Filter |
| GMDH | Group Method of Data Handling |
| WA | Wavelet Analysis |
| R2 | Coefficient of Determination |
| MAPE | Mean Absolute Percent Error |
| RMSE | Root Mean Square Error |
| IMF | Intrinsic Mode Function |
| MAE | Mean Square Error |
| RA | Relative Accuracy |
| BO-MK-RVM | Bayesian optimization MK-RVM |
| VR-BO-MK-RVM | Bayesian optimization MK-RVM of Voltage Recovery Model |

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
