# Peer review of "Remaining Useful Life Prediction Method of PEM Fuel Cells Based on a Hybrid Model"

_electronics, doi:10.3390/electronics12183883_

Round 1

Reviewer 1 Report (Previous Reviewer 1)

The authors have appropriately modified the manuscript according to the reviewers' suggestions

Author Response

Reviewer 2 Report (Previous Reviewer 2)

The introduction is well stated and gives the sufficient background for the easy comprehension of the status of development of the PEM fuel cells technology.  Moreover, relevant previous studies available in literature are cited.

·       -   I will suggest introducing the methodology using for the tests before the section 2 ‘data set section’ in order to help the reader understanding what the steps are the Authors follows to achieve the results.

·        -  Line 352 – please revise the typo error.

·     - Results stated at line 134-135 ‘the output voltage of the reactor has an obvious negative correlation with the time’ should be supported by data.

·        -  Section '3. Discuss' should be improved or deleted because it is very similar to the conclusion section and the information provided in this section do not provide any added value.

Author Response

Reviewer 3 Report (Previous Reviewer 3)

The reviewed article deals with the issue of Remaining Useful Life Prediction Method of PEM Fuel Cell 2 Based on Hybrid Model. The article is very interesting, however, can the EMD analysis be replaced by another? e.g. using SVD or wavelets?. A comparative analysis would be very interesting.

Author Response

This manuscript is a resubmission of an earlier submission. The following is a list of the peer review reports and author responses from that submission.

Round 1

Reviewer 1 Report

This manuscript presents a prediction method based on the voltage recovery model and Bayesian optimization of a multi-kernel relevance vector machine by using a set of data from proton-exchange membrane fuel cells.

To predict the remaining useful life of proton-exchange membrane fuel cells, different models were used: the empirical mode decomposition method, the multi-kernel relevance vector machine, the Bayesian optimization algorithm, and the voltage recovery model.

This method has great practical value because it deals with the long-term prediction of the remaining useful life of fuel cells.

The manuscript is well written. The Introduction and Conclusion sections are clear and concise. The manuscript is publishable in Electronics. Maybe, the authors should consider the following minor points:

- FCLAB should be a better described. It seems like an acronym.

- Keywords: It should be PEM Fuel Cell, such as in the Title, instead of PEMFC.

- Line 130 on page 4: The format is missing not only in the text but also in the equations 3-5.

- Figure 3: In the titles, the separation between the magnitude and the unit should be spaced, such as in Figure 4.

- Equations 13-14: Revise the format.

Reviewer 2 Report

This paper proposes a prediction method based on the voltage recovery model and Bayesian optimization of a multi-kernel relevance vector machine (MK-RVM) To predict the remaining useful life (RUL) of proton exchange membrane fuel cell (PEMFC) in advance.

Proton exchange membrane fuel cell (PEMFC) is considered as a promising energy source for variety of energy-conversion technologies such as electric vehicles and stationary power stations due to its advantages of the low pollution and noise and high-power generation efficiency. The performance degradation while is PEMFC is operating is still a big issue in this field as it is an inevitable effect.  Consequently, the estimation of remaining useful life (RUL) a crucial problem that needs to be studied to assure the stability and security of PEMFC.

  • The introduction and the state of art in the field of PEM Fuel Cell should be improved analysing previous additional relevant studies carried out by other authors and presenting the current development state of this technology.
  • Methodology that has been applied by the authors is properly presented but the results achieved are not clear, they should be discussed in a more rigours way in order to demonstrate to the reader the significant contribution that this work can give in PEMFC field.
  • In the same way, the section discussion should be deeply improve as in the current state does not contribute to the scientific soundness of the paper. Finally, the conclusions should be improved by analyzing and summarizing the main quantity and qualitative results achieved through the experiments carried out. 
  • Additional references should be included to support the work. 

Reviewer 3 Report

The bibliography is biased. Contains almost 100% citations of scientists from China. I understand that no one else in the world deals with similar topics? What is the reason for the policy of quoting only authors from China???

Reviewer 4 Report

The study presented is related with an important topic, but the presentation of the ideas could be clearer and more objective. For example saying  that a PEMFC has a " high energy conversion efficiency" is relative; in Fig 2 some variables are not immediately identified and should be; in line 138 the function r(t) is used twice  with different meanings; in line 194, when referring a start-stop, the exact steps of this procedure should be described;  the time intervals defined at the beginning of paragraph do not have a clear justification; when saying that the method is 95% effective, some comparison with other methods should be done.

The text is readable, but some points must be corrected, like the use of upper/lower case, some spelling punctuation, some spacing between lines and the  text in point 6 of line 190

Round 2

Reviewer 2 Report

Thank you for your efforts in revising the work.

Reviewer 3 Report

The article is not submitted in an MDPI template.